# Research

Subject Areas:
energy

Keywords:
chainsaw arm, narrow pillars, roof cutting, stress release, gob-side roadway

Author for correspondence:
Bin Yu
e-mail: yubin0352@163.com

# Research on stress release for the gob-side roadway using the roof-cutting technology with a chainsaw arm

Yang Tai[1], Bin Yu[2], Binwei Xia[3], Zhao Li[2] and Hongchu Xia[4]

[1]School of Mines, China University of Mining and Technology, Xuzhou 221116, People's Republic of China
[2]Datong Coal Mine Group Co. Ltd., Datong 037000, People's Republic of China
[3]State Key Laboratory of Coal Mine Disaster Dynamics and Control, Chongqing University, Chongqing 400044, People's Republic of China
[4]College of Architectural Engineering, Dalian University, Liaoning, Dalian 116622, People's Republic of China

YT, 0000-0002-9969-4860; BY, 0000-0003-3763-873X

The narrow pillar mining method is widely adopted for working faces in coal mines. However, in cases of an overlying hard roof, a suspended triangle roof plate or a cantilever will be formed near the goaf. At this point, the coal pillar extrusion and serious deformation will occur in the gob-side roadway. In order to mitigate the problem, the roof-cutting technology with a chainsaw arm and its equipment have been developed. In this paper, based on the analysis of deformation and failure characteristics of 2312 roadway, which is close to the goaf of 2311 working face in Tashan Coal, the roof-cutting technology with a chainsaw arm was chosen to be applied in 2311 roadway. Then, the roof-cutting process and the load acting on the coal pillar were discussed and analysed. A numerical model was established to analyse the stress releasing effects after roof cutting. Moreover, the roof-cutting height and the support parameters of the roadway were optimized through numerical analysis and the results manifested that the roof cutting was the most effective when the roof-cutting height was 6.4 m. After roof cutting, the vertical stresses within the coal pillars were lowered by about 25.0%. Finally, the roof-cutting experiment was carried out in the 2311 roadway in Tashan Coal Mine. The on-site roof-cutting depth was 6.4 m and the roof-cutting width was 42 mm guided by the numerical analysis. To verify the stress-relieving effects, the borehole stress meters were applied to monitor the peak advancing stresses of narrow pillars at various depths. The

measured results indicated that the peak advancing stresses decreased by 22.8% on average, and therefore, roof cutting and stress releasing effects were achieved.

# 1. Introduction

In order to isolate the mined-out areas of adjacent working faces, the coal pillars with a certain width are usually reserved [1,2]. However, the coal pillars will directly lead to the waste of mine resources and reduce the service life of coal mines. Under such a circumstance, to reduce coal loss and improve coal recovery rate, narrow pillar mining technology has been developed to extend coal mine service life [3]. The protective coal pillar size has also been reduced from 20–40 to 3–6 m width [4,5], therefore, the resource recovery rate will increase [6]. Also, the gob-side roadway will be easier to maintain owing to the stress decreasing zone of the surrounding rock. On the benefits mentioned above, the narrow pillar mining technology has been widely applied.

The core problem to be considered during the narrow pillar mining method is the coal pillar size and coal pillar stability. To determine the reasonable size of narrow pillars, a lot of research has been conducted by physical simulation, numerical simulation and theoretical calculation methods. For example, the influences of pillar size on the deformation of the roadway surrounding rock and stress were studied by the physical simulation [7]. The effects of coal pillar size on fracture propagation in narrow pillars, pillar stress and deformation were studied by UDEC Trigon logic and FLAC software [8]. The rational pillar size was given by the width of the pillar's plastic zone based on beam theory. For pillar stability, Singh *et al*. [1] analysed the effects of surrounding rock stress on pillar stability during the mining process by the theoretical calculation; Renani *et al*. [9] analysed the failure laws of the pillar owing to size and shape; Elmo *et al*. [10] analysed the influence of joints and fractures on the pillar stability by an integrated numerical modelling–discrete fracture network approach; Zhou *et al*. [11] studied the dynamic process of coal pillar deformation and failure by the physical simulation method; and Bertuzzi *et al*. [12] gave the strength calculation equation of the coal pillar by the empirical formula, and the numerical simulation was used for verification.

The aforementioned studies mainly focus on the size and stability of narrow pillars. However, if a suspended triangle roof plate or a cantilever is formed near the goaf, at this point, the gob-side coal pillar and roadway will undergo serious deformation. In this case, the hydraulic fracturing method is usually adopted to cut the hard roof. But it is widely accepted that hydraulic fracturing is hard to control hydraulic fracture propagation. Therefore, a continuous, accurate and mechanized roof-cutting method is urgently needed to cut the hard roof precisely. After many years' research, the roof-cutting method with chainsaw arm and its equipment has been developed by the Datong Coal Mine Group. In this paper, the 2312 roadway deformation and failure characteristics in Tashan Coal Mine were firstly analysed. Then, the load acting on the coal pillar before and after roof cutting were given. Next, the stress releasing effects were analysed by the numerical simulation method, and the key parameters of the roof-cutting technology were optimized. Finally, the industrial experiment was carried out in the 2311 roadway in Tashan Coal Mine.

# 2. The engineering background

## 2.1. Geological conditions

As shown in figure 1*a*, Tashan Coal Mine is located in the Datong coalfield of Datong city, Shanxi province, China. The 4# coal seam in the 8311 working face is now under exploitation. The thickness of the 4# coal seam is between 3.20 and 3.60 m with an average of 3.40 m. The buried depth was between 350 and 500 m. In order to ensure normal and continuous mining, and enhance the recovery rate of coal resources, the excavation of the 2312 roadway of the 8312 working face has been completed before the mining of the 8311 working face, and 6.0 m narrow pillars are set between the 8311 and 8312 working face. There is intact, hard and thick rock above the 8311 working face. Figure 1*b* shows the specific occurrence conditions of the roof.

## 2.2. Roadway deformation and failure characteristics

After the mining of the 8311 working face in Tashan Coal Mine, a suspended triangle roof plate or a cantilever was formed in the roadway near the goaf of the 8311 working face. At this point, the coal

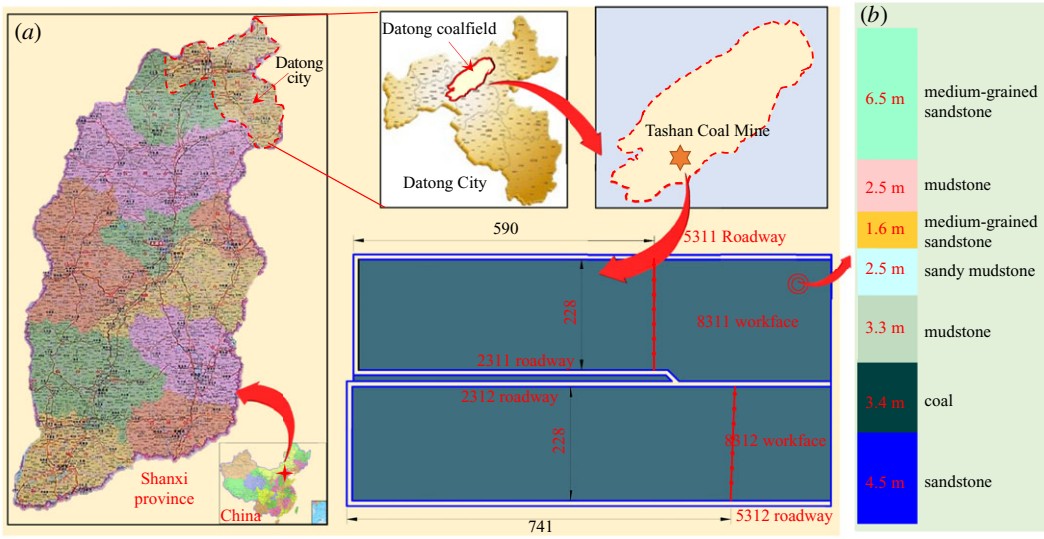

**Figure 1.** Geological conditions. (*a*) The workface conditions; (*b*) drill columns.

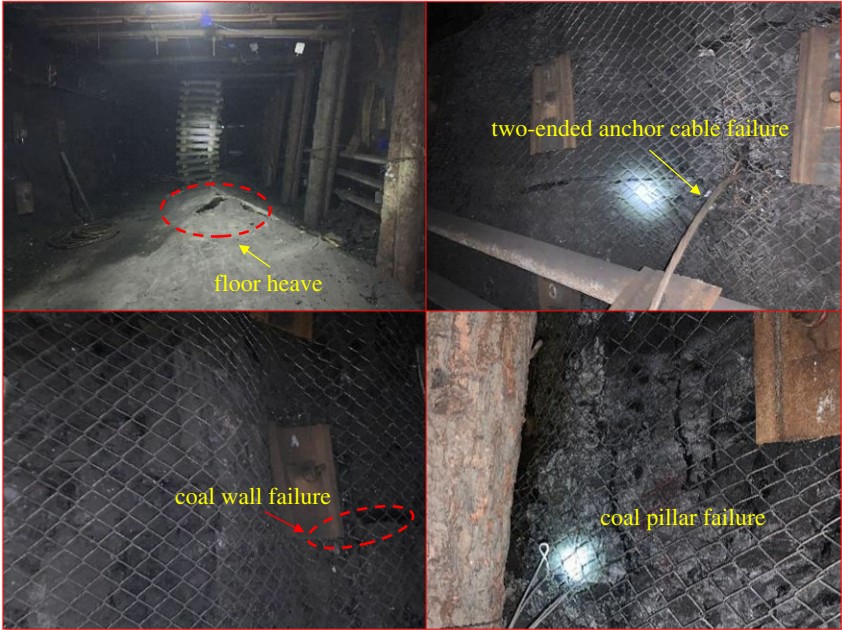

**Figure 2.** Roadway deformation and failure.

pillars extruded and serious deformation and failure have been noticed in the 2312 roadway, as shown in figure 2. Strong strata behaviours, including floor heave, two-ended anchor cable failure, the coal wall failure and coal pillar failure, have taken place in the roadway. Therefore, the roof-cutting technology is necessary to release stress in the roof. Under such a background, Datong Coal Mine Group has developed the roof-cutting technology with a chainsaw arm.

# 3. The equipment and roof-cutting process for roof-cutting technology with a chainsaw arm

One of the effective methods to control the roadway deformation and failure is to cut the cantilever or the triangular plate by the chainsaw arm. This section mainly introduces the structural composition of the chainsaw arm cutting machine and the roof-cutting process.

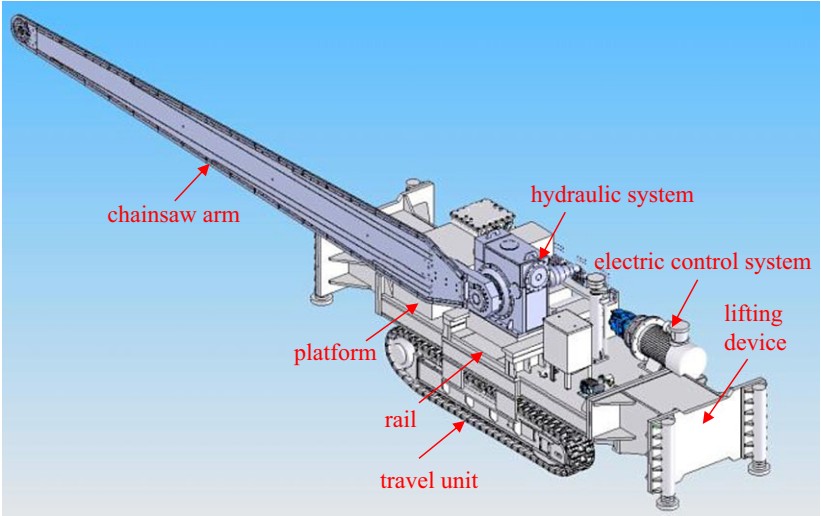

**Figure 3.** The chainsaw arm cutting machine.

## 3.1. The structural composition of the chainsaw arm cutting machine

The chainsaw arm cutting machine is the core equipment of roof cutting. It mainly consists of the platform, chainsaw arm, travel unit, the hydraulic system, electric control system, rail and lifting device. Figure 3 shows its main components.

## 3.2. Roof-cutting process

As shown in figure 4a, the chainsaw arm cutting machine was set in the roadway. In addition, temporary support equipment was applied to control the roadway deformation after the roof-cutting process.

The specific roof-cutting process is stated as follows: (i) to start the travel unit and arrange the cutting machine at the end of the roadway; (ii) to use the travel unit to make the saw close to the coal wall near the goaf and keep the chainsaw arm parallel to the roadway; (iii) to extend the cylinder of the lifting device and make the travel unit get off the ground at a distance of 10–20 mm; (iv) as shown in figure 4b, to start the power head and drive the high-speed rotation of the cutting chain; to move the chainsaw arm from one side to the other side and make it rotate uniformly with the rotation angle of 180°; (v) to rapidly move the chainsaw arm to the original horizontal position when it reaches the other side; (vi) to use the travel unit to move the chainsaw arm cutting machine forward for some distance; and (vii) to make the saw accurately align with the last cutting crack by the rail. Then to repeat steps (ii)–(vi) and start the next roof cutting. Figure 4c shows the specific process.

# 4. The loads acting on the narrow pillars before and after roof cutting

## 4.1. The loads acting on the narrow pillars before roof cutting

Before the roof cutting, the broken positions of the basic roof are usually outside of the coal pillar, right above the roadway and inside the coal wall. For the three positions, the loads acting on the narrow pillars have been given [13], as shown in table 1.

## 4.2. The load acting on the narrow pillars after roof cutting

Compared with the traditional working face, the broken position of the working face after roof cutting was at the boundary of the goaf. The load acting on the coal pillars after roof cutting were given [13], as shown in table 2.

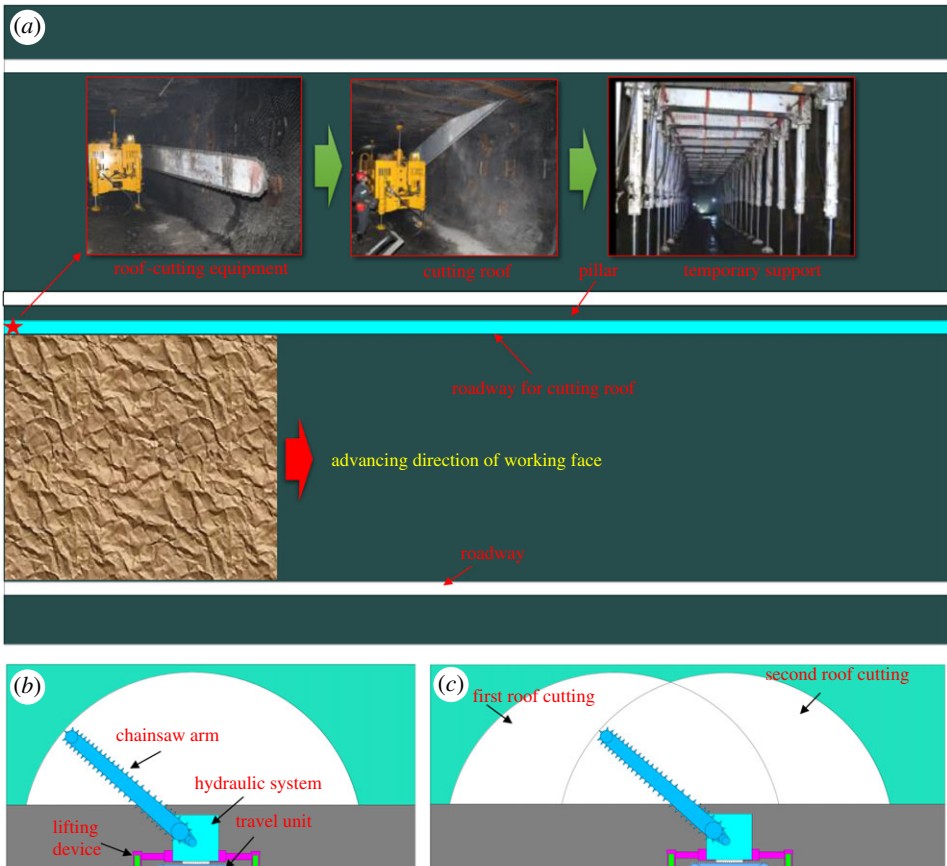

**Figure 4.** Roof-cutting process.

# 5. The stress releasing effects for the gob-side roadway by the roof-cutting technology with a chainsaw arm

## 5.1. The numerical model of the gob-side roadway using the roof-cutting technology

### 5.1.1. Strength reduction for coal and rock mass at the stope

Table 3 shows the mechanical parameters of the intact coal and rock mass obtained in the laboratory experiment. There are a lot of natural and irregular joints and fractures in the coal and rock mass, so the strength of the coal and rock mass in the laboratory is generally higher than that at the stope. In order to simulate the strength reduction, Hoek and Brown proposed the reduction factors of $m_b$, $s$ and $a$, whose formulae are shown as follows [14]:

$$m_b = m_i \exp\left(\frac{\text{GSI} - 100}{28 - 14D}\right), \tag{5.1}$$

$$s = \exp\left(\frac{\text{GSI} - 100}{9 - 3D}\right) \tag{5.2}$$

and

$$a = \frac{1}{2} + \frac{1}{6}(e^{-\text{GSI}/15} - e^{-20/3}), \tag{5.3}$$

where $m_b$ is a reduced value (for the rock mass) of the material constant $m_i$ (for the intact rock); $s$ and $a$ are constants which depend upon the characteristics of the rock mass; GSI is the geological strength index; and parameter $D$ is a 'disturbance factor' which depends upon the degree of disturbance to which the rock mass has been subjected by blast damage and stress relaxation. It varies from 0 for

**Table 1.** The load acting on the narrow pillars before the roof cutting. (P.S.: $\Delta S_B$ is the subsidence of the rock block B at B' (m); $\Delta S_C$ is the subsidence of the rock block C at C' (m); $T_B$ and $T_C$ are the lateral horizontal thrusts of the rock block B at B' and the rock block C at C' (kN); $N_B$ and $N_C$ are shear stresses of the rock block B and C (kN); $\sigma$ is the support of the coal in the plastic zone to the roof (MPa); $q$ is the weight of the basic roof and the average load of the overlying soft strata (kN m$^{-1}$); $q_0$ is the average load of the immediate roof (kN m$^{-1}$); $M_A$ and $M_B$ are the residual moments of the rock beam B at A' and B' (kN·m); $M_0$ is the bending moment of the immediate roof to the basic roof (kN·m); $h$ is the thickness of the roof (m); $L$ is the lateral fracture span with the basic roof; $P_1$ is the load acted on the coal pillar with the broken line of the basic roof at the side of the goaf (kN); $x_0$ is the width of limit equilibrium zone of the lateral coal mass (m); $a$ is the width of the roadway retained (m); $b$ is the width of coal pillar (m); $P_2$ is the load acted on the coal pillar with the fracture line right above the roadway (kN); $d'$ is the horizontal distance between the edge of the coal pillar and the fracture line of the roof (m); $P_3$ is the load acted on the coal pillar with the fracture line at the side of the coal mass around the roadway (kN).)

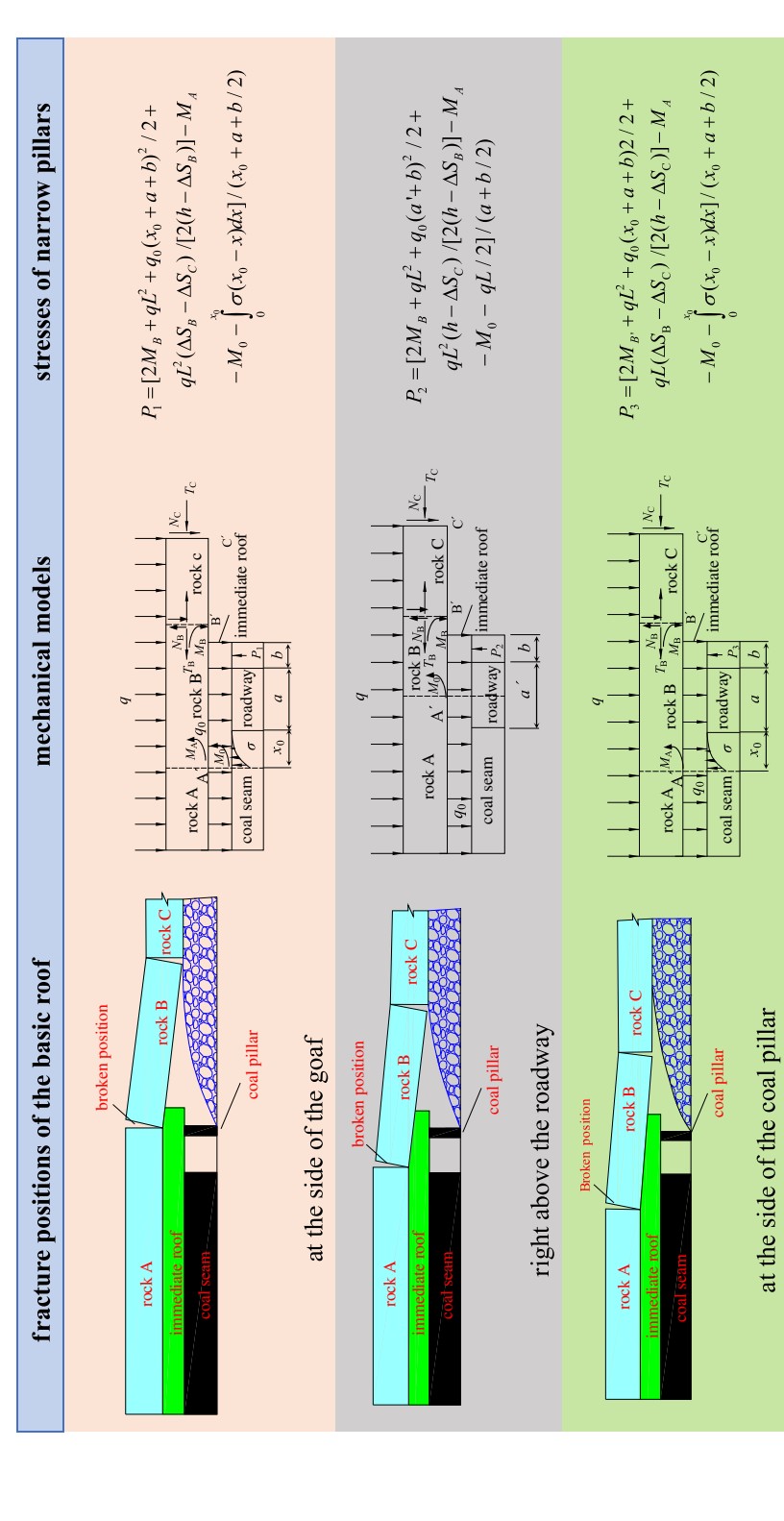

| fracture positions of the basic roof | mechanical models | stresses of narrow pillars |
|---|---|---|
| at the side of the goaf | | $P_1 = [2M_B + qL^2 + q_0(x_0 + a + b)^2 / 2 + qL^2(\Delta S_B - \Delta S_C) / [2(h - \Delta S_B)] - M_A - M_0 - \int_0^{x_0} \sigma(x_0 - x)dx] / (x_0 + a + b / 2)$ |
| right above the roadway | | $P_2 = [2M_B + qL^2 + q_0(a' + b)^2 / 2 + qL^2(h - \Delta S_C) / [2(h - \Delta S_B)] - M_A - M_0 - qL / 2] / (a + b / 2)$ |
| at the side of the coal pillar | | $P_3 = [2M_{B'} + qL^2 + q_0(x_0 + a + b)2 / 2 + qL(\Delta S_B - \Delta S_C) / [2(h - \Delta S_C)] - M_A - M_0 - \int_0^{x_0} \sigma(x_0 - x)dx] / (x_0 + a + b / 2)$ |

**Table 2.** The load acting on the narrow pillars after roof cutting [13]. (P.S.: $\Delta S_C$ is the subsidence of the rock block C at C′ (m); $T_B$ and $T_C$ are the lateral horizontal thrusts of the rock block B at B′ and the rock block C at C′ (kN); $N_B$ and $N_C$ are shear stresses of the rock blocks B and C, respectively (kN); $\sigma$ is the support of the coal mass in the plastic zone to the roof (MPa); $q$ is the weight of the basic roof and the average loads of the overlying soft strata (kN m$^{-1}$); $q_0$ is the average loads the immediate roof (kN m$^{-1}$); $M_A$ and $M_B$ are the residual moments of the rock beam B at A′ and B′ (kN·m); $M_0$ is the bending moment of the immediate roof to the basic roof (kN·m); $K_G$ is the support coefficient of the gangue in the goaf (kN m$^{-1}$); $F_G$ is the support stress of the goaf to the roof (kN m$^{-1}$); $h_0$ is the thickness of the basic roof (m); $P$ is the load acted on the coal pillar (kN); $x_0$ is the width of the limit equilibrium zone of the lateral coal mass (m); $a$ is the width of the roadway retained (m); $b$ is the width of the coal pillar (m).)

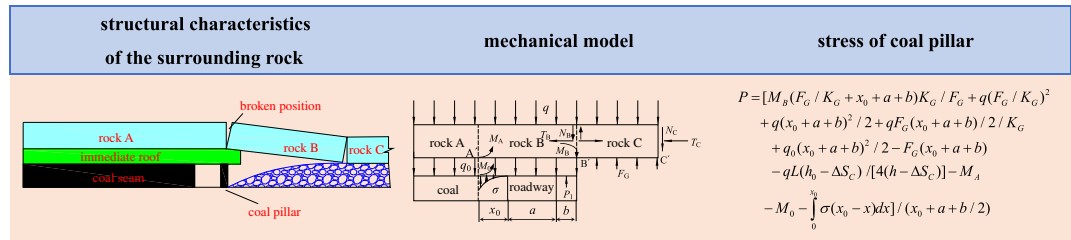

| structural characteristics of the surrounding rock | mechanical model | stress of coal pillar |
|---|---|---|

$$P = [M_B(F_G/K_G + x_0 + a + b)K_G/F_G + q(F_G/K_G)^2$$
$$+ q(x_0 + a + b)^2/2 + qF_G(x_0 + a + b)/2/K_G$$
$$+ q_0(x_0 + a + b)^2/2 - F_G(x_0 + a + b)$$
$$- qL(h_0 - \Delta S_C)/[4(h - \Delta S_C)] - M_A$$
$$- M_0 - \int_0^{x_0} \sigma(x_0 - x)dx]/(x_0 + a + b/2)$$

undisturbed *in situ* rock masses to 1 for very disturbed rock masses. Here, it is 0 from [15]; $m_i$ is a material constant for the intact rock [16]:

$$\sigma_{cm} = \sigma_{ci}\frac{(m_b + 4s - a(m_b - 8s))(m_b/4 + s)^{as-1}}{2(1+a)(2+a)} \tag{5.4}$$

and

$$E_{cm} = \left(1 - \frac{D}{2}\right)\sqrt{\frac{\sigma_{ci}}{100}}10^{GSI-10/40}, \tag{5.5}$$

where $\sigma_{ci}$ is the uniaxial compressive strength (UCS) of the intact rock pieces, $\sigma_{cm}$ and $E_{cm}$ are the reduced uniaxial compressive strength and elastic modulus in the stope, respectively.

RocData software provided the empirical parameters of GSI and empirical values of rock mass $m_i$ of different lithology. $m_b$, $s$ and $a$ of different strata could be calculated by RocData, as shown in table 3.

### 5.1.2. The failure criterion of the coal and rock mass

Mohr–Coulomb failure criterion is generally recognized as the constitutive model of the coal and rock mass. The failure criterion may be represented in the plane ($\sigma_1$, $\sigma_3$), as illustrated in figure 5. The failure envelope from point A to B could be expressed by Mohr–Coulomb yield function, as follows [17,18]:

$$f^s = \sigma_1 - \sigma_3 N_\varphi + 2c\sqrt{N_\varphi}, \tag{5.6}$$

where $\varphi$ is the internal friction angle, $c$ is the cohesion, and $N_\varphi = (1 + \sin \varphi)/(1 - \sin \varphi)$.

The failure envelope from point B to C could be expressed by Mohr–Coulomb yield function, as follows [19]:

$$f^t = \sigma_{ci} - \sigma_3. \tag{5.7}$$

### 5.1.3. The constitutive model of the caving zone

The working face under the traditional caving method needs to get the height of the caving zone at the stope. The regression equation of the height of the caving zone has been given as follows [20,21]:

$$H_0 = \frac{100h}{(c_1h + c_2)}, \tag{5.8}$$

where $h$ is the mining height (m); $c_1$ and $c_2$ are parameters related to the roof lithology, as shown in table 4.

According to equation (5.8) and the type of the immediate roof, the height of the caving zone was 6.4 m. Then, the caving zone could be obtained by the roof-cutting technology and the caving angle of the goaf.

**Table 3.** Mechanical parameters of the rock and coal mass.

| lithology | intact rock | | | | | | | | | | |
| | elastic modulus E (GPa) | $\sigma_{ci}$ (MPa) | Poisson ratio $\mu$ | cohesion c (MPa) | internal friction angle (°) | GSI | $m_i$ | $m_b$ | s | a |
|---|---|---|---|---|---|---|---|---|---|---|
| medium-grained sandstone | 32.3 | 52.1 | 0.26 | 12.3 | 21.3 | 85 | 17 | 9.95 | 0.189 | 0.50 |
| mudstone | 28.5 | 75.1 | 0.26 | 7.8 | 24.1 | 79 | 10 | 4.72 | 0.97 | 0.50 |
| medium-grained sandstone | 29.3 | 85.2 | 0.25 | 9.6 | 21.3 | 91 | 18 | 13.05 | 0.37 | 0.50 |
| sandy mudstone | 33.8 | 74.3 | 0.26 | 10.8 | 23.3 | 86 | 16 | 9.70 | 0.21 | 0.50 |
| mudstone | 30.1 | 19.1 | 0.30 | 8.3 | 24.1 | 90 | 9 | 6.30 | 0.33 | 0.50 |
| coal seam | 22.3 | 19.2 | 0.26 | 7.8 | 21.2 | 85 | 10 | 5.85 | 0.19 | 0.50 |
| sandstone | 35.3 | 51.3 | 0.28 | 13.2 | 19.2 | 86 | 11 | 6.67 | 0.21 | 0.50 |

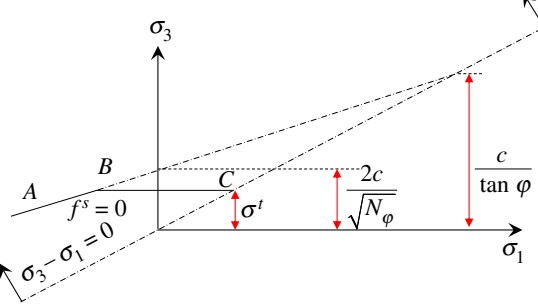

**Figure 5.** Mohr–Coulomb failure criterion.

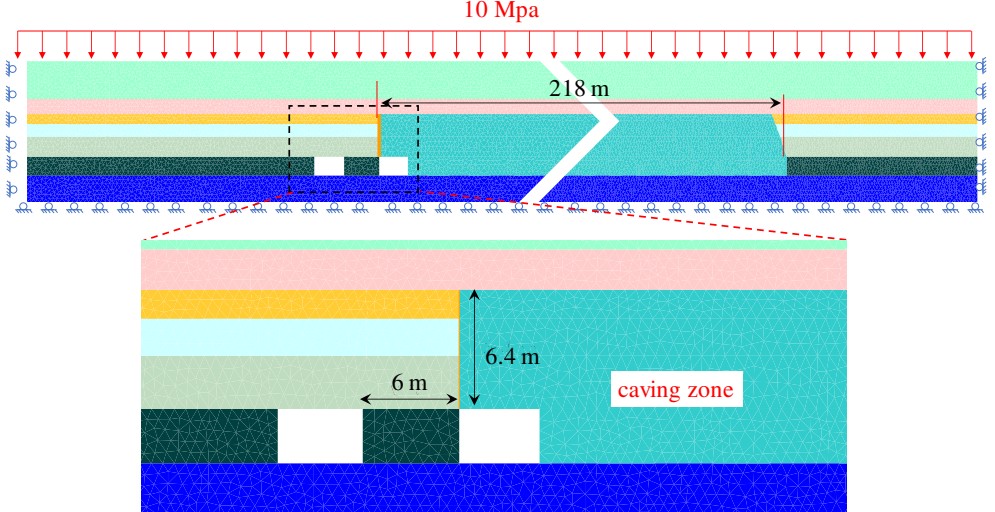

**Figure 6.** The numerical model.

**Table 4.** Coefficients of the height of the caving height [22].

| | | coefficients | |
| --- | --- | --- | --- |
| type of the immediate roof | UCS (MPa) | $c_1$ | $c_2$ |
| hard | >40 | 2.1 | 16 |
| medium hard | 20–40 | 4.7 | 19 |
| soft | <20 | 6.2 | 32 |

To use the finite-element software to simulate the compaction characteristics of the rock mass in the caving zone, Salamon put forward the compaction theory of broken rock mass in the caving zone, that is, the stress–strain relation of the rock mass could be obtained. The stress–strain relation was given as follows [23]:

$$\sigma_{\mathrm{v}} = \frac{E_0 \varepsilon}{(1 - \varepsilon/\varepsilon_{\mathrm{m}})}, \tag{5.9}$$

where $\sigma_{\mathrm{v}}$ is the vertical stress of the goaf (MPa); $E_0$ is the initial tangent modulus of the rock mass in the caving zone (MPa); $\varepsilon$ is the current vertical strain; $\varepsilon_{\mathrm{m}}$ is the largest vertical strain.

$\varepsilon_{\mathrm{m}}$ could be fixed by the following equation [21]:

$$\varepsilon_{\mathrm{m}} = \frac{b - 1}{b}, \tag{5.10}$$

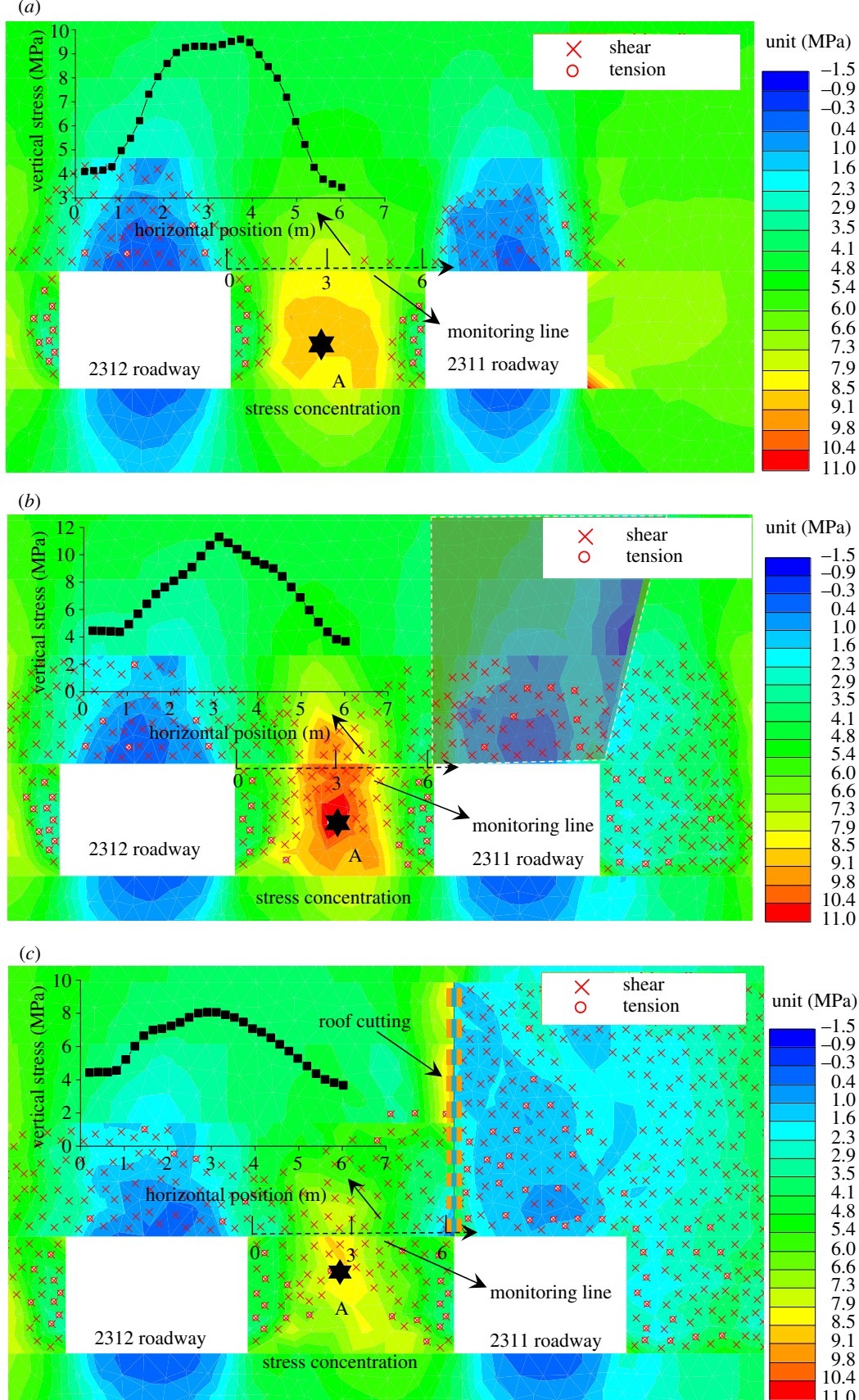

**Figure 7.** Coal pillar stresses. (*a*) After the roadway excavation; (*b*) after mining without roof cutting and (*c*) after mining with roof cutting.

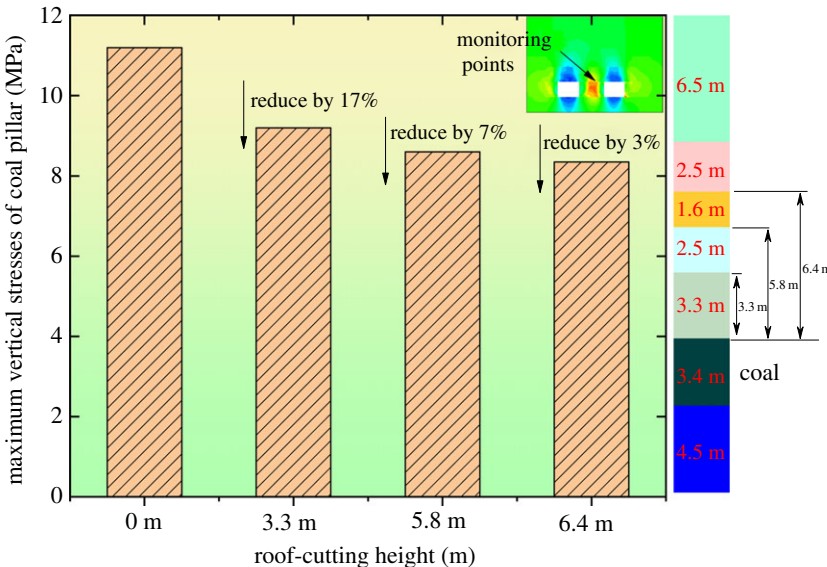

**Figure 8.** The maximum vertical stresses of the coal pillar under different roof-cutting heights.

where $b$ is the comprehensive bulking coefficient of the rock mass in the caving zone. It could be fixed by the following equation [20]:

$$b = 1 + 0.01(c_1 h + c_2). \tag{5.11}$$

$E_0$ could be calculated by the following equation [21]:

$$E_0 = 10.39\sigma_{ci}^{1.042}/b^{7.7}. \tag{5.12}$$

Through equations (5.10) and (5.11), the maximum vertical strain $\varepsilon_m$ and the comprehensive bulking coefficient $b$ are 0.35 and 1.54, respectively. The stress–strain relation of the compacted rock mass could be obtained by substituting $\varepsilon_m$ and $b$ into equation (5.9).

### 5.1.4. The numerical model

As shown in figure 6, a two-dimensional plane model was established with the length of 343.3 m and the height of 24.3 m according to the geological conditions of the 8311 working face in Tashan Coal Mine. The grid size of the model was between 0.5 and 1.0 m [24]. The Mohr–Coulomb failure criterion was used for the rock and coal mass [25]. The double-yield model was used for the caving zone [21]. The right and left sides of the model was restricted its horizontal displacement and the bottom of the model was restricted its vertical displacement. A 10 MPa vertical load was applied on the top to replace the weight of the overlying 400 m rock strata, and the ground stress was also applied [26,27].

## 5.2. The stress releasing effects for the gob-side roadway

The stress release for the roadway was able to reduce the pillar stress and failure, so the roof-cutting effects were evaluated from those two aspects. The vertical stresses and the failure areas of surrounding rock after the roadway excavation, after mining without the roof cutting and after mining with the roof cutting is shown in figure 7. Meanwhile, the vertical stresses on top of narrow pillars were extracted.

(i) As shown in figure 7a, the coal pillar experienced failure to some extent after the roadway excavation, and the elastic area of the central position still existed. The roadway deformation could be controlled by setting the anchorage section of the anchor bolt in the elastic area. The vertical stress of the coal pillar showed the single peak, that is, the position of the coal pillar stress concentration was in the elastic core. The maximum vertical stress was 9.6 MPa, and the minimum vertical stress was 4.1 MPa.

(ii) As shown in figure 7b, basically the plastic failure domains the coal pillar failure mode after mining without roof cutting, and the bearing capacity of the coal pillar was reduced. The vertical stress presented a sharp peak. The maximum vertical stress was 11.2 MPa, and the minimum amount was 4.3 MPa.

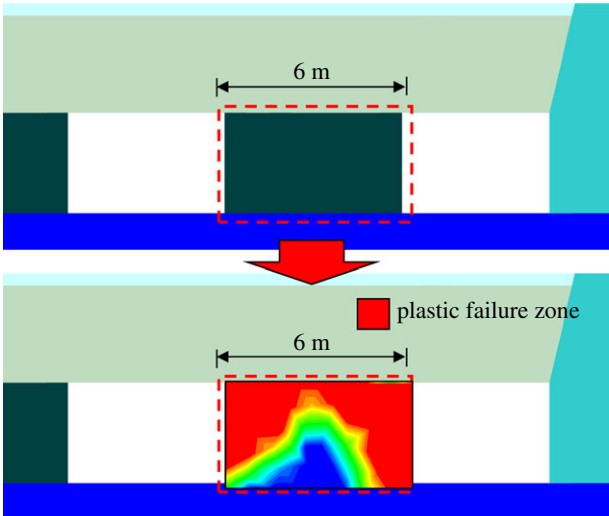

**Figure 9.** Failure laws of the coal pillar.

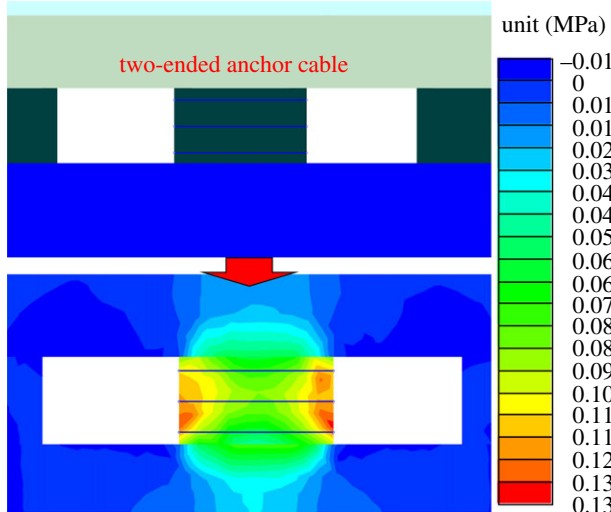

**Figure 10.** The additional stress field of the two-ended anchor cable.

(iii) As shown in figure 7c, the plastic failure after the mining with roof cutting, and the bearing capacity of the narrow pillar was also reduced. The vertical stress presented a single peak. The maximum and minimum vertical stresses were 8.1 and 4.3 MPa, respectively.

(iv) Compared with the maximum vertical stress after the mining without roof cutting, that of the coal pillar after the mining with roof cutting changed from 11.2 to 8.4 MPa, decreasing by 25.0%. It was indicated that the roof cutting could reduce the vertical stress of the coal pillar to a certain degree, thus achieving the stress releasing effects for the gob-side roadway.

# 6. The key parameters for roof-cutting technology and roadway support

The key parameters of roof cutting include the roof-cutting height. Therefore, the reasonable roof-cutting height should be determined first, and then, the roadway support parameters should be optimized.

## 6.1. The roof-cutting height

The roof-cutting height directly determines the stress releasing effects. In theory, the higher the roof-cutting height, the better the roof cutting and stress releasing effects, and the higher the costs. Therefore, it is necessary to choose a proper height. As shown in figure 8, the numerical simulation

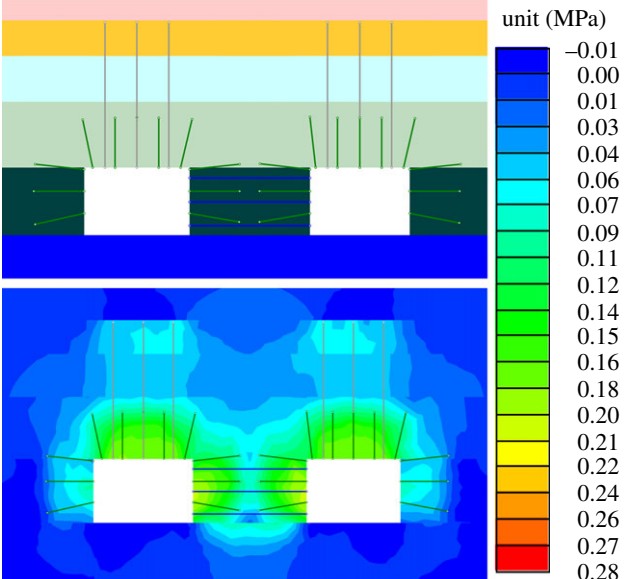

**Figure 11.** The additional stress field formed by the anchor bolt and cable.

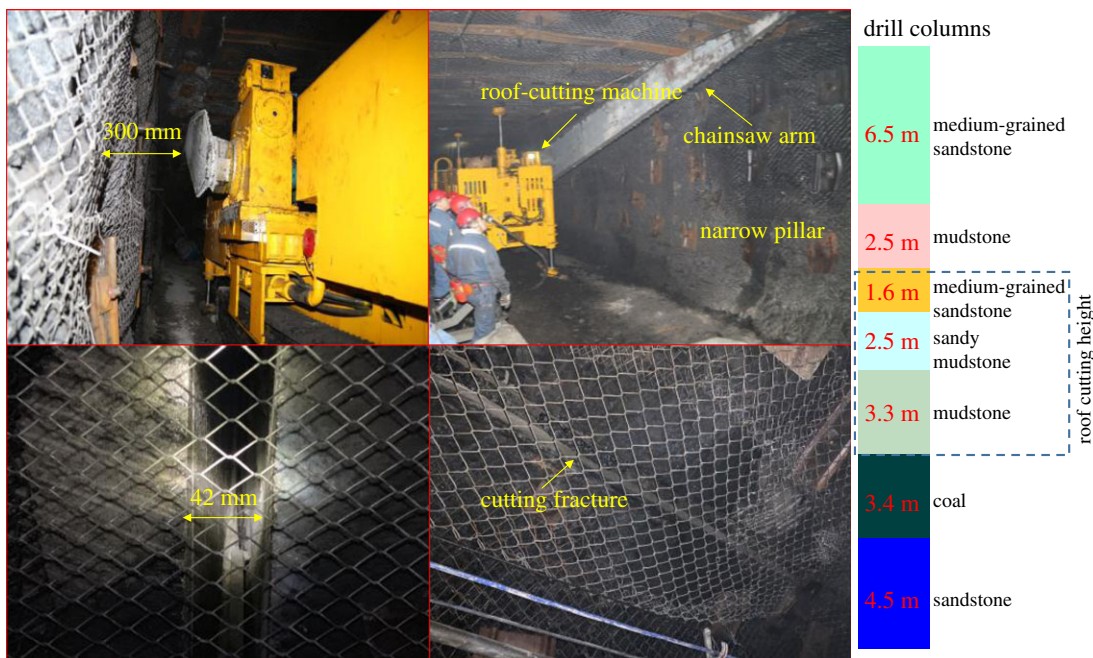

**Figure 12.** The chain arm sawing roof.

method was used to extract the maximum vertical stresses of the coal pillar under different roof-cutting heights. Through comparison and analysis, the rational roof-cutting height was determined.

As shown in figure 8, when the roof-cutting heights were 0, 3.3, 5.8 and 6.4 m, the maximum stresses were 11.2, 9.2, 8.6 and 8.4 MPa, respectively. This indicates that, as the height increased, the maximum stresses decreased constantly. When the height was 6.4 m, the maximum stresses remained basically unchanged, so the roof-cutting height was fixed as 6.4 m.

## 6.2. Roadway support parameters

### 6.2.1. Two-ended anchor cable

The numerical analysis implied that the plastic failure occurred in the top region of the coal pillar after the advance of the working face, as shown in figure 9. Therefore, there was no elasticity on the top of the coal

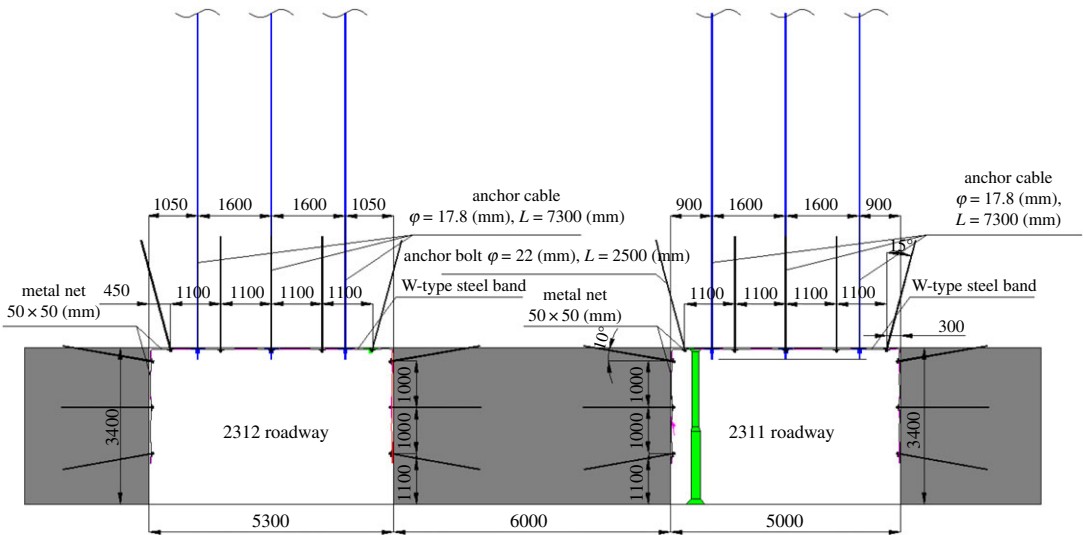

**Figure 13.** Roadway support parameters.

**Table 5.** Technical parameters of the roof-cutting machine with a chainsaw arm.

| item | parameters |
|---|---|
| type | KLJ7000 |
| total length (m) | 11.2 |
| total width (m) | 1.82 |
| total height (m) | 2.86 |
| machine weight (t) | about 27 |
| ground clearance (mm) | 288 |
| swing angle of the chain arm (°) | 180 |
| cutting speed of the chain arm (° h$^{-1}$) | 0–90 |
| adaptive slope (°) | ±12 |
| cutting hardness (MPa) | ≤8 |
| total power (kW) | 75 |
| standard cutting depth (m) | 7.0 |
| actual cutting depth (m) | 6.45 |
| comprehensive noise value during the excavation dB(A) | 85 |

pillar, which makes it difficult to provide an anchorage zone for the anchor bolt or anchor cable. Thus, in order to mitigate further damage, the two-ended anchor cable was applied for reinforced support.

One of the key methods to control the further damage of the coal pillar was to select the appropriate parameters of the two-ended anchor cable. The length of the two-ended anchor cable was the width of the coal pillar. At the same time, 0.3 m was reserved on two sides of the coal pillar for rigging installation, so the length was 6.6 m. According to the pre-stressed criterion, a complete pressure layer needed to be formed on two sides of the coal pillar [28]. Finally, through continuous trials, the length of the anchor cable was 6.6 m, the pre-tightening force was 150 kN and the spacing was 1200 × 2400 mm. Figure 10 shows the additional stress field by using the aforementioned parameters.

### 6.2.2. Parameters of the anchor bolt and anchor cable

The parameters of the anchor bolt and cable were also determined according to the pre-stressed criterion. To get proper parameters, the matching between the anchor bolt and cable also should be taken into consideration at the same time. The length of the bolt was determined as 2.4 m, the spacing was

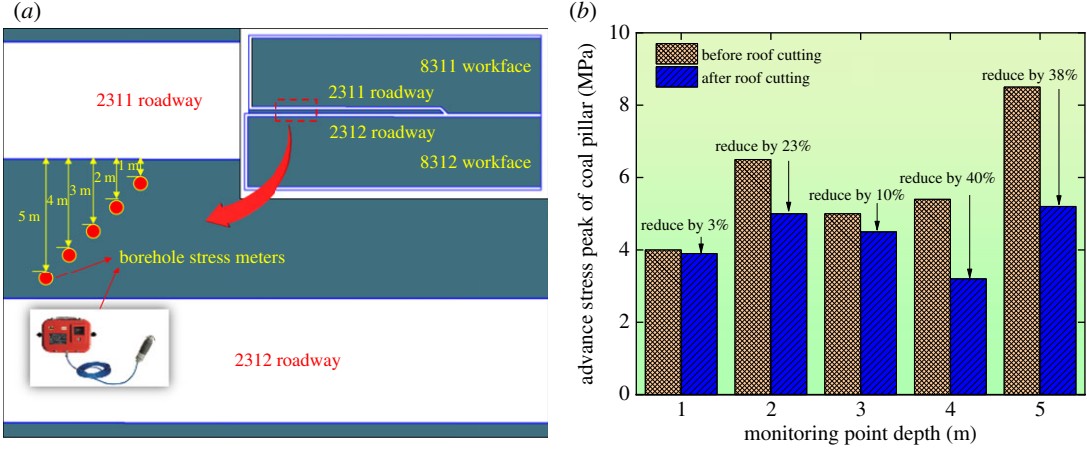

**Figure 14.** The advancing stress peak of the coal pillar. (*a*) The monitoring scheme and (*b*) the monitoring results.

1100 mm × 1200 mm and the pre-tightening force was 100 kN. The length of the anchor cable was 7.3 m, the spacing was 1600 mm × 3600 mm and the pre-tightening force was 150 kN. Figure 11 shows the corresponding additional stress field.

# 7. The engineering application

## 7.1. Roof-cutting parameters

The KLJ7000 roof-cutting machine with a chainsaw arm was set in the 2311 mining roadway. As shown in figure 12, the roof cutting was carried out 300 mm away from the coal wall with the roof-cutting height of 6.4 m and the roof-cutting width of 42 mm. The cutting fracture penetrated the mudstone layer, sandy mudstone layer and medium-grained sandstone layer within the roof. Table 5 shows the specific technical parameters of the cutting machine.

## 7.2. Support parameters

Figure 13 shows the support of the 2311 roadway of the 8311 working face in Tashan Coal Mine.

The roof support parameters are described as follows: the left-hand screw-thread steel bolt without longitudinal reinforcement was used. It had a length of 2.4 m, with a diameter of 22 m. The spacing is 1100 mm × 1200 mm, and the pre-tightening force 100 kN. The tray specification is 130 mm × 130 mm × 10 mm. The anchor cable had a length of 7.3 m, with a spacing of 1600 mm × 3600 mm. The pre-tightening force is 150 kN and the tray specification is 300 × 300 × 16 mm. The roof used the diamond metal net pattern with the size of 50 × 50 mm.

The single hydraulic props with articulated beam and metal shoe were adopted in the range of 20 m advance working face. A row of concentrated probes was set on the side of 2311 roadway for advanced support with the row spacing of 1.0 m, 300 mm away from the coal pillar.

## 7.3. The advancing stress peak of the coal pillar

The stress state of the coal pillar could directly present the stress releasing effects. As shown in figure 14*a*, the borehole stress meters were arranged to monitor the stress states at depths of 1, 2, 3, 4 and 5 m. In order to compare the stress states before and after roof cutting, the advancing stress peak of the narrow pillar was given during the mining process, as shown in figure 14*b*.

Figure 14*b* indicated that: (i) after roof cutting, all vertical stresses of the coal pillar at various depths showed different degrees of reduction; (ii) the extents of the stress reduction were the largest at the pillar depths of 4 and 5 m, decreasing by 38% and 40%, respectively. The stresses increased by 10% and 23% at the pillar depths of 2 and 3 m, respectively. At the depth of 1 m, the pressure releasing effects of the coal pillar were not significant; (iii) the average stress of the coal pillar decreased by 22.8%, which was basically consistent with the numerical simulation result of 25.0%. It was proved that the result of the numerical simulation was reliable.

# 8. Conclusion

(i) The roof-cutting equipment and process were firstly introduced. Moreover, the load acting on the coal pillars before and after the roof cutting are compared.

(ii) The numerical simulation method was adopted to analyse the stress releasing effects for the gob-side roadway. Moreover, the roof-cutting height and support parameters of the roadway were optimized.

(iii) The roof-cutting experiment was carried out in the 2311 roadway of no. 3 panel in the Tashan Coal Mine. The on-site roof-cutting depth was 6.4 m and the roof-cutting width was 42 mm. The measured results indicated that the maximum vertical stresses of the coal pillar decreased after the roof cutting, and thus, roof cutting and stress releasing effects were achieved.

Permission to carry out fieldwork. We obtained the appropriate permissions and licenses to conduct the fieldwork detailed in our study.

Data accessibility. Data is available from the Dryad Digital Repository: https://doi.org/10.5061/dryad.8cz8w9gkt [29].

Authors' contributions. Y.T. participated in the design of the study and drafted the manuscript; H.X. carried out the statistical analyses; Z.L. collected field data; B.Y. conceived of the study and designed the study; B.X. coordinated the study and helped draft the manuscript. All authors gave final approval for publication.

Competing interests. The authors declare no competing financial interests.

Funding. This work was supported by the Outstanding Innovation Scholarship for Doctoral Candidate of CUMT (2019YCBS005).

Acknowledgements. The authors gratefully acknowledge the financial support from the organization above.

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
