## [Reviewer comments · Royal Society Open Science]

Review History

RSOS-191663.R0 (Original submission)

Review form: Reviewer 1

Is the manuscript scientifically sound in its present form?

Yes

Are the interpretations and conclusions justified by the results?

Yes

Is the language acceptable?

Yes

Do you have any ethical concerns with this paper?

No

Have you any concerns about statistical analyses in this paper?

No

Recommendation?

Accept as is

Comments to the Author(s)

In the narrow pillar mining technology, the deformation and failure of the free roadway is one of the common problems in the coal mine. In this paper, Research on Stress Release for Gob-Side Roadway Using Roof-Cutting Technology with Chainsaw Arm, the roof-cutting technology with chainsaw arm can solve the above problems well and has outstanding engineering value. At the same time, the key of roof cutting is optimized by theoretical derivation, numerical calculation and engineering test. The structure of the paper is reasonable, the chart processing is exquisite and the discussion is detailed, which reflects the author's good scientific research ability.

Therefore, this paper suggests that the paper should be accepted after revision. At the same time, the paper also has the following small problems:

- 1) There are some punctuation problems in some parts of the article, such as page 2 line 46. Floor weave, two ended anchor cable failure, the coal wall failure, coal pillar failure and etc take place in the roadway.
- 2) Why the numerical simulation is adopted after the theoretical calculation in this paper. What is the relationship between the both method? Is it mutual verification or theoretical calculation has some shortcomings? Please give an explanation.
- 3) There are some unclear expressions about formula (4) and (5) in this paper. Please give some explanations. Are σ cm and E_{cm} the corresponding reduced uniaxial compressive strength and elastic modulus in the stope?
- 4) In Page5 Line 29, the author uses load to replace the effect of overlying surrounding rock on the working face. Is this simplification reasonable? Is there any reference to use this method? Please give a detailed explanation.
- 5) In Page 5 Line 36-38, the author's statement is not accurate. Please check it carefully and give corresponding modification.
- 6) Whether the chainsaw arm machine can cut the roof at a certain angle, and whether the rock collapse angle is considered, please give some explanation.

Review form: Reviewer 2

Is the manuscript scientifically sound in its present form?

Yes

Are the interpretations and conclusions justified by the results?

Yes

Is the language acceptable?

Yes

Do you have any ethical concerns with this paper?

No

Have you any concerns about statistical analyses in this paper?

No

Recommendation?

Accept with minor revision (please list in comments)

Comments to the Author(s)

The narrow pillar mining technology in this paper "Research on Stress Release for Gob-Side Roadway Using Roof-Cutting Technology with Chainsaw Arm" is one of the commonly used mining methods in China, and the problem of gob-side roadway described in this paper is also a prominent problem in the narrow pillar mining technology. The roof-cutting technology with

chainsaw arm proposed by the author can effectively form continuous cutting cracks and effectively solve the problem of coal pillar pressure relief. In this paper, the numerical calculation method is mainly used to give the rule of stress change of coal pillar before and after pressure relief, and optimize the roof-cutting height and anchor cable support parameters. On the whole, the paper structure is reasonable and detailed, which is a good academic paper. Therefore, I suggest that the article can be accepted after revision. At the same time, there are also the following problems that the authors need to consider:

1. In Chapter 5.1.1 of this paper, the author describes the relationship between the stope rock mass and the rock mass in the experiment, but mechanical parameters needed to reduce should be given, firstly. Is whether the mechanical parameters correspond to the formula (4) - (5).
2. The roof-cutting technology with chainsaw arm is an innovative technology. Does it also have other engineering application or application prospect in the field of coal mine, such as floor pressure relief or non-pillar mining technology?
3. The author gives the reasonable roof-cutting height and the corresponding supporting parameters. These parameters are aimed at the special geological conditions of Tashan Coal Mine of the same coal group. Are they worth popularizing and applying?
4. It is an important parameter that coal pillar on the boundary in numerical simulation. Please give the specific value in the paper and explain the basis of selecting the value?
5. Figure 7 shows the vertical stresses and the failure areas of surrounding rock after the roadway excavation, after mining without roof cutting and after mining with the roof cutting. Meanwhile, the vertical stresses on top of narrow pillars were extracted. There are some inaccuracies in the author's statement. Please check it carefully and give corresponding modifications.
6. Hoek Brown criterion is a common rock criterion in recent years. The value of D is a core parameter affecting rock strength. The author chooses 0 here, and please give a reason.

Decision letter (RSOS-191663.R0)

21-Jan-2020

Dear Dr Tai

On behalf of the Editors, I am pleased to inform you that your Manuscript RSOS-191663 entitled "Research on Stress Release for Gob-Side Roadway Using Roof-Cutting Technology with Chainsaw Arm" has been accepted for publication in Royal Society Open Science subject to minor revision in accordance with the referee suggestions. Please find the referees' comments at the end of this email.

The reviewers and handling editors have recommended publication, but also suggest some minor revisions to your manuscript. Therefore, I invite you to respond to the comments and revise your manuscript.

- Ethics statement

- Data accessibility

It is a condition of publication that all supporting data are made available either as supplementary information or preferably in a suitable permanent repository. The data accessibility section should state where the article's supporting data can be accessed. This section should also include details, where possible of where to access other relevant research materials

such as statistical tools, protocols, software etc can be accessed. If the data has been deposited in an external repository this section should list the database, accession number and link to the DOI for all data from the article that has been made publicly available. Data sets that have been deposited in an external repository and have a DOI should also be appropriately cited in the manuscript and included in the reference list.

If you wish to submit your supporting data or code to Dryad (<http://datadryad.org/>), or modify your current submission to dryad, please use the following link:
<http://datadryad.org/submit?journalID=RSOS&manu=RSOS-191663>

- **Competing interests**

- **Authors' contributions**

- **Acknowledgements**

- **Funding statement**

Because the schedule for publication is very tight, it is a condition of publication that you submit the revised version of your manuscript before 30-Jan-2020. Please note that the revision deadline will expire at 00.00am on this date. If you do not think you will be able to meet this date please let me know immediately.

When submitting your revised manuscript, you will be able to respond to the comments made by

the referees and upload a file "Response to Referees" in "Section 6 - File Upload". You can use this to document any changes you make to the original manuscript. In order to expedite the processing of the revised manuscript, please be as specific as possible in your response to the referees. We strongly recommend uploading two versions of your revised manuscript:

If your manuscript is newly submitted and subsequently accepted for publication, you will be asked to pay the article processing charge, unless you request a waiver and this is approved by Royal Society Publishing. You can find out more about the charges at <https://royalsocietypublishing.org/rsos/charges>. Should you have any queries, please contact openscience@royalsociety.org.

on behalf of the Associate Editor, and Professor R. Kerry Rowe (Subject Editor)
openscience@royalsociety.org

Associate Editor Comments to Author:

Thank you for submitting your manuscript to Royal Society Open Science. As you can see in the reports below, both referees are supportive of publication, but recommend revisions to your paper. Please ensure to answer these thoroughly within your point-by-point response letter. Please also provide a tracked changes version of your manuscript document, which highlights the revisions you have made. We look forward to receiving your revised paper.

Reviewer comments to Author:

Reviewer: 1

Comments to the Author(s)

In the narrow pillar mining technology, the deformation and failure of the free roadway is one of the common problems in the coal mine. In this paper, Research on Stress Release for Gob-Side Roadway Using Roof-Cutting Technology with Chainsaw Arm, the roof-cutting technology with chainsaw arm can solve the above problems well and has outstanding engineering value. At the same time, the key of roof cutting is optimized by theoretical derivation, numerical calculation and engineering test. The structure of the paper is reasonable, the chart processing is exquisite and the discussion is detailed, which reflects the author's good scientific research ability. Therefore, this paper suggests that the paper should be accepted after revision. At the same time, the paper also has the following small problems:

- 1) There are some punctuation problems in some parts of the article, such as page 2 line 46. Floor weave, two ended anchor cable failure, the coal wall failure, coal pillar failure and etc take place in the roadway.
- 2) Why the numerical simulation is adopted after the theoretical calculation in this paper. What is the relationship between the both method? Is it mutual verification or theoretical calculation has some shortcomings? Please give an explanation.
- 3) There are some unclear expressions about formula (4) and (5) in this paper. Please give some explanations. Are σ cm and E_{cm} the corresponding reduced uniaxial compressive strength and elastic modulus in the stope?
- 4) In Page5 Line 29, the author uses load to replace the effect of overlying surrounding rock on the working face. Is this simplification reasonable? Is there any reference to use this method? Please give a detailed explanation.
- 5) In Page 5 Line 36-38, the author's statement is not accurate. Please check it carefully and give corresponding modification.
- 6) Whether the chainsaw arm machine can cut the roof at a certain angle, and whether the rock collapse angle is considered, please give some explanation.

Reviewer: 2

Comments to the Author(s)

The narrow pillar mining technology in this paper "Research on Stress Release for Gob-Side

Roadway Using Roof-Cutting Technology with Chainsaw Arm" is one of the commonly used mining methods in China, and the problem of gob-side roadway described in this paper is also a prominent problem in the narrow pillar mining technology. The roof-cutting technology with chainsaw arm proposed by the author can effectively form continuous cutting cracks and effectively solve the problem of coal pillar pressure relief. In this paper, the numerical calculation method is mainly used to give the rule of stress change of coal pillar before and after pressure relief, and optimize the roof-cutting height and anchor cable support parameters. On the whole, the paper structure is reasonable and detailed, which is a good academic paper. Therefore, I suggest that the article can be accepted after revision. At the same time, there are also the following problems that the authors need to consider:

1. In Chapter 5.1.1 of this paper, the author describes the relationship between the stope rock mass and the rock mass in the experiment, but mechanical parameters needed to reduce should be given, firstly. Is whether the mechanical parameters correspond to the formula (4) - (5).
2. The roof-cutting technology with chainsaw arm is an innovative technology. Does it also have other engineering application or application prospect in the field of coal mine, such as floor pressure relief or non-pillar mining technology?
3. The author gives the reasonable roof-cutting height and the corresponding supporting parameters. These parameters are aimed at the special geological conditions of Tashan Coal Mine of the same coal group. Are they worth popularizing and applying?
4. It is an important parameter that coal pillar on the boundary in numerical simulation. Please give the specific value in the paper and explain the basis of selecting the value?
5. Figure 7 shows the vertical stresses and the failure areas of surrounding rock after the roadway excavation, after mining without roof cutting and after mining with the roof cutting. Meanwhile, the vertical stresses on top of narrow pillars were extracted. There are some inaccuracies in the author's statement. Please check it carefully and give corresponding modifications.
6. Hoek Brown criterion is a common rock criterion in recent years. The value of D is a core parameter affecting rock strength. The author chooses 0 here, and please give a reason.

Author's Response to Decision Letter for (RSOS-191663.R0)

See Appendix A.

Decision letter (RSOS-191663.R1)

03-Feb-2020

Dear Dr Tai,

It is a pleasure to accept your manuscript entitled "Research on Stress Release for Gob-Side Roadway Using Roof-Cutting Technology with Chainsaw Arm" in its current form for publication in Royal Society Open Science. The comments of the reviewer(s) who reviewed your manuscript are included at the foot of this letter.

on behalf of Prof R. Kerry Rowe (Subject Editor)
openscience@royalsociety.org

Appendix A

Cover letter

Dear editor

Thank you for giving me the chance to revise my manuscript. All revisions are marked with red and blue fonts in the article. If you have any questions, please contact me.

Yours sincerely,

Bin Yu

E-mail: cumtcqty@163.com

Telephone number: +86-15105205502

Institution: Datong Coal Mine Group Co. Ltd., Datong, 037000, China.

The Revision for Comments

Reviewer: 1

1. There are some punctuation problems in some parts of the article, such as page 2 line 46. Floor heave, two ended anchor cable failure, the coal wall failure, coal pillar failure and etc take place in the roadway.

Thank you for your comments. The article is modified as follows:

floor heave, two-ended anchor cable failure, the coal wall failure, coal pillar failure and etc.

2. Why the numerical simulation is adopted after the theoretical calculation in this paper. What is the relationship between the both method? Is its mutual verification or theoretical calculation having some shortcomings? Please give an explanation.

In this paper, the theoretical calculation can only give the average load of the coal pillar before and after the roof cutting, but not the maximum stress of the coal pillar, so there are some defects in the theoretical calculation. In this paper, numerical simulation is used to make up for the problems in theoretical calculation, so the two are complementary.

3. There are some unclear expressions about formula (4) and (5) in this paper. Please give some explanations. Are σ_{cm} and E_{cm} the corresponding reduced uniaxial compressive strength and elastic modulus in the stope?

The paper adds the explanation of σ_{cm} and E_{cm} as follows:

σ_{cm} and E_{cm} are the reduced uniaxial compressive strength and elastic modulus in the stope, respectively.

4. In Page5 Line 29, the author uses load to replace the effect of overlying surrounding rock on the working face. Is this simplification reasonable? Is there any reference to use this method? Please give a detailed explanation.

In the numerical simulation, the strata above the bending zone can be replaced by loads. This simplification is reflected in many articles, as follows:

1 Barton, N. & Pandey, S.K. 2011 Numerical modelling of two stopping methods in two Indian mines using degradation of c and mobilization of ϕ based on Q-parameters. International Journal of Rock Mechanics and Mining Sciences 48, 1095-1112. (doi: 10.1016/j.ijrmms.2011.07.002).

2 Lin, H., Xiong, Z.Y., Liu, T.Y., Cao, R.H. & Cao, P. 2014 Numerical simulations of the effect of bolt inclination on the shear strength of rock joints. International Journal of Rock Mechanics and Mining Sciences 66, 49-56. (doi: 10.1016/j.ijrmms.2013.12.010).

5. In Page 5 Line 36-38, the author's statement is not accurate. Please check it carefully and give corresponding modification.

The article is modified as follows:

It was shown that the vertical stresses and the failure areas of surrounding rock after the roadway excavation, after mining without roof cutting and after mining with the roof cutting in Figure 7. Meanwhile, the vertical stresses on top of narrow pillars were extracted.

6. Whether the chainsaw arm machine can cut the roof at a certain angle, and whether the rock collapse angle is considered, please give some explanation.

The chainsaw arm machine can cut rock at a certain angle, but it needs to be equipped with a certain angle adjusting device. In this paper, the rock collapse angle is not considered, because the roof is cut vertically, the area of the suspended roof is the smallest.

Reviewer: 2

1. In Chapter 5.1.1 of this paper, the author describes the relationship between the stope rock mass and the rock mass in the experiment, but mechanical parameters needed to reduce should be given, firstly. Is whether the mechanical parameters correspond to the formula (4) - (5)?

In this paper, the description of the coefficient to be reduced is added, which is σ_{cm} and E_{cm} of formula (4) - (5), as follows:

σ_{cm} and E_{cm} are the reduced uniaxial compressive strength and elastic modulus in the stope, respectively.

2. The roof-cutting technology with chainsaw arm is an innovative technology. Does it also have other engineering application or application prospect in the field of coal mine, such as floor pressure relief or non-pillar mining technology?

In this paper, some application prospects of chain arm saw cutting are added as follows:

The technology of roof cutting with chainsaw arm is mainly used for cutting rock, which can be used in many fields such as non-pillar mining method, non-pillar mining method with protective layer, pressure relief for roadway floors and so on.

3. The author gives the reasonable roof-cutting height and the corresponding supporting parameters. These parameters are aimed at the special geological conditions of Tashan Coal Mine of the same coal group. Are they worth popularizing and applying?

The parameters in this paper are selected according to the specific engineering background of Tashan Coal Mine, so these parameters may not be applied to other mining areas, but this paper provides a parameter optimization method for the other mining areas.

4. It is an important parameter that coal pillar on the boundary in numerical simulation. Please give the specific value in the paper and explain the basis of selecting the value?

The coal pillar on the boundary in the numerical simulation are 50 m. This parameter is a set of optimal solutions after constant adjustment of boundary protection in numerical simulation.

5. Figure 7 shows the vertical stresses and the failure areas of surrounding rock after the roadway excavation, after mining without roof cutting and after mining with the roof cutting. Meanwhile, the vertical stresses on top of narrow pillars were extracted. There are some inaccuracies in the author's statement. Please check it carefully and give corresponding modifications.

The article is modified as follows:

It was shown that the vertical stresses and the failure areas of surrounding rock after the roadway excavation, after mining without roof cutting and after mining with the roof cutting in Figure 7. Meanwhile, the vertical stresses on top of narrow pillars were extracted.

6. Hoek Brown criterion is a common rock criterion in recent years. The value of D is a core parameter affecting rock strength. The author chooses 0 here, and please give a reason.

In this paper, the value of D is 0 because the roadway is mined by the roadheader rather than by blasting, so the surrounding rock damage of the roadway can be ignored, which can be proved in the following references:

Piotr, M., Łukasz, O. & Piotr, B. 2017 Modelling the Small Throw Fault Effect on the Stability of a Mining Roadway and Its Verification by In Situ Investigation. *Energies* 10. (doi:10.3390/en10122082).